# Redox Homeostasis in Red Blood Cells: From Molecular Mechanisms to Antioxidant Strategies

**DOI:** 10.3390/cimb47080655

**Published:** 2025-08-14

**Authors:** Sara Spinelli, Angela Marino, Alessia Remigante, Rossana Morabito

**Affiliations:** 1Department of Chemical, Biological, Pharmaceutical and Environmental Sciences, University of Messina, 98125 Messina, Italy; sa.spinelli@unime.it (S.S.); marinoa@unime.it (A.M.); rmorabito@unime.it (R.M.); 2Department of Biomedical, Dental and Morphological and Functional Imaging, University of Messina, 98125 Messina, Italy

**Keywords:** red blood cells (RBCs), oxidative stress, reactive oxygen species (ROS), band 3 protein, phosphorylation pathways, RBC cytoskeleton, RBC membrane, hemoglobin (Hb), antioxidants

## Abstract

Red blood cells (RBCs) are uniquely vulnerable to oxidative stress due to their role in O_2_ transport and their high content of heme iron and polyunsaturated fatty acids (PUFAs). Despite lacking nuclei and organelles, RBC homeostasis relies on a finely tuned redox system to preserve membrane integrity, cytoskeletal organization, and metabolic function. Impairment of this delicate balance results in a series of oxidative events that ultimately leads to the premature clearance of RBCs from the bloodstream. This review outlines the main oxidative mechanisms that affect RBC at different levels, such as membrane, cytoskeleton, and intracellular environment, with a focus on the molecular targets of reactive species. The role of major antioxidant systems in preventing or reversing redox damage will also be examined, revealing their multiple mechanisms of action ranging from direct ROS scavenging to the enhancement of endogenous antioxidant defense pathways. Redox regulatory mechanisms in RBCs are required to maintain membrane integrity, cytoskeletal organization, and metabolic function. Disruption of these processes causes several oxidative processes that trigger premature RBC removal. Cumulative evidence places oxidative stress at the core of RBC dysfunction in both physiological aging and pathological conditions, including diabetes, inflammatory conditions, and hemolytic disorders. Antioxidant-based strategies, rather than providing generalized protection, should aim to selectively target the specific molecular pathways affected in distinct clinical settings.

## 1. Introduction

Mature red blood cells (RBCs) are the most abundant cell type in the human body, with an estimated average of 25 trillion per individual [1]. Their primary function is to deliver oxygen (O_2_) to tissues via narrow capillaries with a thickness of 2 µm, as well as to significantly contribute to the regulation of whole-blood acid–base homeostasis [2]. These functions are tightly interlinked, as the O_2_-binding affinity of hemoglobin (Hb), which contains heme-bound ferrous ion (Fe^2+^), is regulated by O_2_ partial pressure, pH, and 2,3-biphosphoglycerate levels [3]. The basic requirements for an effective O_2_ delivery include a specialized membrane structure, an active energy source within the cell and an adequate supply of specific phosphorylated intermediates. Mature RBCs maintain a complex balance of enzymes, proteins, carbohydrates, lipids, and ions, which are necessary for cellular metabolism and functionality. An important outcome of an imbalance among these components is a compromised ability to cope with oxidative stress. Red blood cells are extremely susceptible to oxidative stress, which causes biochemical, topological, and mechanical alterations. This vulnerability derives from three main factors: the chemical composition of the RBC membrane (especially the content of polyunsaturated acids (PUFAs)), the continuous exposure of such cells to O_2_, and the rate of Hb auto-oxidation [4,5]. In addition, RBCs interact dynamically with inflammatory and oxidative mediators produced in other cells, further amplifying their exposure to stressors. These aspects, together with the lack of nucleus and cytoplasmic organelles, make RBCs particularly prone to oxidative stress [3], paving the way to several investigations giving rise to a new term, “erythropathy” [6], accounting for the abnormal oxidative state associated with different pathological conditions of RBCs [7,8,9]. Among oxidant stressors, reactive oxygen species (ROS) deserve much attention. They, rather than representing a single entity, constitute a broad and heterogeneous group of chemically reactive molecules with a wide range of biological functions [3,10]. The complex biology and redox physiology of RBCs may be framed within the concept of the reactive species “interactome”, a dynamic oxidation–reduction network consisting of chemical interactions between reactive species and their cellular targets. In this context, Hb and its various redox forms exert a central role, ranging from reduced forms (oxyHb and deoxyHb) to the oxidized one (methemoglobin (metHb)) [11]. Specifically, endogenous and exogenous ROS induce iron oxidation in Hb, thus converting Fe^2+^-containing Hb into Fe^3+^-containing metHb. The formation of Fe^3+^ promotes the production of iron-dependent free radicals through the Fenton reaction:(1)Fe^2+^ + H_2_O_2_ → Fe^3+^ + OH^•^ + OH^−^ which causes lipid peroxidation, hemolysis, and endothelial disruption [12]. This reaction is triggered by oxidative stress conditions, as excessive amounts of O_2_ may induce the release of iron from Hb and ferritin (Figure 1). Hydroxyl radical (OH^•^) is an incredibly reactive non-specific free radical that rapidly interacts with nearby targets, such as proteins and lipids, within a nanometer radius of its origin [13]. The extremely rapid oxidative reactions mediated by OH^•^ contribute to its short intracellular half-life, which is approximately 10^−9^ s. Compared to other ROS produced in cells, it is considered the most dangerous, since no enzymatic defense system can neutralize it [3,12]. Beyond iron, the protein moiety of Hb could also contribute to ROS formation under specific conditions. Normally, Fe^3+^-metHb is converted back to its reduced form by cytochrome b5 reductase, according to the following reaction:(2)Fe^3+^-metHb + NADH + H^+^ → Fe^2+^-Hb + NAD^+^

However, under sustained oxidative stress conditions, the amount of NADH reducing equivalent declines. In this context, the heme group of Hb undergoes further degradation into quaternary compounds (i.e., hemin, hemo-chromogen derivates, and heme degradation products), thus resulting in ROS production [14,15]. The redox cycle is further maintained by the Haber–Weiss reaction, which reduces Fe^3+^ from the Fenton reaction to Fe^2+^:(3)Fe^3+^ + O_2_^•−^ → Fe^2+^ + O_2_

Superoxide anion (O_2_^•−^) is produced when O_2_ accepts an electron to complete its orbital, resulting in an unpaired electron and a net negative charge [16]. In RBCs, the auto-oxidation of oxyHb can lead to the formation of both O_2_^•−^ and metHb:(4)Fe^2+^-Hb + O_2_ → Fe^3+^-metHb + O_2_^•−^

The production of O_2_^•−^ could be elicited by the oxidative-dependent activation of NADPH oxidase [17], according to the reaction:(5)NADPH + 2O_2_ → NADP^+^ + 2 O_2_^•−^ + H^+^

The aberrant activation of NADPH oxidase may result in impaired cell deformability, metHb formation, and decreased cell survival. The activity of this enzyme is regulated by complex phosphorylation mechanisms that influence ROS production and significantly contribute to cellular oxidative stress. A key player in this process is protein kinase C (PKC), a serine/threonine kinase that, once activated, phosphorylates specific NADPH oxidase subunits, thereby promoting the assembly of the active enzyme complex on the RBC membrane [17]. Superoxide anion is spontaneously or enzymatically converted by superoxide dismutase (SOD) into hydrogen peroxide (H_2_O_2_), a more stable product [3]. As an oxidizing compound, H_2_O_2_ exhibits high specificity towards protein cysteine thiols and Fe-S clusters. Even at low concentrations (10 µM), H_2_O_2_ may cause oxidative injury to cells. At higher concentrations, H_2_O_2_ inactivates enzymes involved in cellular metabolism, such as glyceraldehyde-3-phosphate dehydrogenase [18], thus resulting in ATP depletion. Despite this, RBCs rely on an efficient endogenous antioxidant system to counteract the detrimental effects of H_2_O_2_, involving catalase (CAT), glutathione peroxidase (GpX), and peroxiredoxins (Prx) [19]. The activation of these enzymes depends on H_2_O_2_ concentrations: GpX and Prx are stimulated at low levels (up to 5 µM), while CAT is activated at higher levels. Consequently, GpX and Prx represent the first defense line against H_2_O_2_ originating from endogenous sources (e.g., Hb auto-oxidation), whereas CAT operates under sustained oxidative stress conditions derived from exogenous sources [20,21]. Alongside the endogenous production of reactive species, exogenous sources play a pivotal role in amplifying oxidative stress impact on RBCs. ROS produced in other body districts and released into the bloodstream can be stored within RBCs [12]. Specifically, systemic pathological conditions such as diabetes, obesity, and cardiovascular diseases can trigger neutrophil and macrophage activation, resulting in the release of ROS into the plasma [22,23]. Similarly, vascular endothelial cells may release ROS as a result of vascular dysfunction, hyperglycemia or hypertension. Furthermore, environmental and dietary stressors, e.g., toxins, xenobiotics, oxidizing drugs, radiation, and pollutants, contribute to systemic oxidative burden, with ROS accumulation within RBCs (Figure 1) [3,24].

Oxidative stress induces profound structural changes in RBCs, resulting in impaired RBC function, metabolic dysregulation, and cellular molecular inactivation [12]. The accumulation of ROS triggers pathophysiological mechanisms involving several cellular pathways, such as lipid peroxidation, membrane and cytoskeletal protein oxidation, and Heinz body formation, namely denatured Hb aggregates that bind to the membrane, triggering RBC splenic removal [25,26]. Despite their destructive potential, moderate concentrations of ROS act as second messengers in physiological processes, including cell signaling, maintenance of redox balance, and clearance of necrotic or apoptotic cells [3,4]. A hallmark consequence of oxidative stress is eryptosis, a type of RBC suicidal death characterized by a massive Ca^2+^ influx and the activation of Gardos channels, calpain, and scramblase [27], resulting in cell shrinkage, vesicle formation, and phosphatidylserine (PS) externalization. The exposure of PS causes the activation of splenic macrophages, which phagocytose and degrade eryptotic RBCs [28,29]. While eryptosis prevents the release of hemolysis-mediated toxic products into the bloodstream, an excessive eryptosis rate leads to the onset of anemia, which reduces systemic oxygen supply [4]. Therefore, pro-eryptotic and anti-eryptotic mechanisms, as well as the underlying oxidative stress levels, must be adequately balanced [4,30]. The maintenance of RBC redox status is crucial not only to ensure an adequate oxygen delivery to tissues but also to preserve a healthy circulatory system through the interactions of RBCs with other blood cells and the vascular endothelium [31,32]. In this context, exogenous antioxidants introduced through the diet play a pivotal role in enhancing RBC antioxidant defense. The efficacy of an antioxidant molecule depends on its intrinsic properties and its capacity to interact with specific cellular targets. Certain molecules exert their antioxidant activity by directly scavenging ROS, modulating the activity of endogenous antioxidant enzymes, or chelating pro-oxidant metal ions [33,34]. Compounds such as flavonoids, melatonin, and phenolic derivatives have shown substantial efficacy in preserving RBC structural and functional integrity in vitro and in animal models [10,35,36,37]. In light of this, the aim of the present collection is to examine the relationship between oxidative stress and RBC activity, underscoring the importance of antioxidant molecule-mediated redox regulation in maintaining cellular function and structural integrity. Specifically, we explore the molecular mechanisms underlying oxidative damage at the level of three distinct structures of RBCs: plasma membrane, cytoskeleton, and intracellular environment.

## 2. Oxidative Stress at Plasma Membrane Level: Molecular Targets and Underlying Mechanisms

### 2.1. Structure and Functions of the RBC Membrane

#### 2.1.1. Lipid Composition of the RBC Membrane

The lipid composition of the RBC membrane represents one of the most intriguing and functionally critical aspects of its structure. It reflects a complex asymmetric organization that contributes directly to membrane mechanics, biochemical signaling, and the regulation of interactions with the circulatory microenvironment [38]. The lipid bilayer is primarily composed of phospholipids, cholesterol, and, to a lesser extent, glycolipids [39]. Phospholipids are asymmetrically distributed between the inner and outer leaflets of the bilayer, with the phosphatidylcholine and sphingomyelin predominantly in the outer leaflet, and phosphatidylethanolamine, PS, and phosphatidylinositol, enriched in the inner one [40]. This asymmetry is not merely structural; it plays a fundamental physiological role in macrophage recognition and elimination of damaged or senescent RBCs, which expose PS. Therefore, the confinement of PS in the inner layer is essential to prevent a premature removal [40,41]. Under physiological conditions, membrane asymmetry is maintained by a dynamic balance between the activities of three major classes of enzymes: flippases, floppases, and scramblases. Flippases, members of the P4-ATPase family, selectively catalyze the ATP-dependent translocation of PS and phosphatidylethanolamine from the outer to the inner leaflet of the lipid bilayer. In contrast, floppases, that belong to the ATP-binding cassette (ABC) transporter family, promote the ATP-dependent transport of phosphatidylcholine and sphingomyelin from the cytoplasmic leaflet to the extracellular leaflet [42]. Floppases exhibit lower substrate specificity than flippases and are also involved in the regulation of lipid distribution during membrane remodeling or in response to mechanical and oxidative stress. Scramblases, by contrast, mediate a non-selective, bidirectional redistribution of phospholipids between the two membrane leaflets. Unlike flippases and floppases, their activity is energy-independent and becomes prominent under conditions of elevated intracellular Ca^2+^ levels or ATP depletion, hallmarks of oxidative stress, eryptosis, or cellular senescence [43]. Cholesterol, in an almost equimolar ratio to phospholipids, intercalates between the hydrophobic acyl chains of the bilayer, modulating both membrane fluidity and structural organization. At lower temperatures, it increases membrane rigidity, whereas at physiological temperatures it enhances fluidity. Cholesterol also contributes to the formation of ordered microdomains, known as lipid rafts, that are enriched in specific transmembrane proteins and play key roles in molecular clustering and signal transduction [44]. Although they constitute a minor component of the lipid bilayer, glycolipids are restricted to the outer leaflet and fulfil essential roles in immunological recognition, interactions with circulating lectins and parasites such as *Plasmodium falciparum*, and in defining RBC antigenic identity, particularly in the ABO blood group system [45].

#### 2.1.2. Protein Composition of the RBC Membrane

The protein component of the RBC membrane, which accounts for approximately 52% of its dry weight, is broadly categorized into integral and peripheral membrane proteins [46]. Integral proteins span the lipid bilayer and are involved in essential functions such as ion transport, cytoskeletal anchoring, and maintenance of membrane structural integrity [47]. Among these, band 3 (encoded by the *SLC4A1* gene) is the most abundant and functionally significant: it mediates Cl^−^/HCO_3_^−^ anion exchange, thereby enabling efficient CO_2_ transport in the form of bicarbonate and supporting the blood buffering capacity [48]. Band 3 exists as a mixture of dimers and tetramers within the membrane [48]. Structurally, band 3 consists of 14 transmembrane segments organized in two major domains of similar size but distinct functions. The C-terminal domain includes 12–14 transmembrane helices and a short cytoplasmic tail with a carbonic anhydrase binding site and mediates the anion transport function [49]. The N-terminal cytosolic domain is water-soluble and critical for structural roles. It serves as an anchoring platform for proteins such as ankyrin, protein 4.1, protein 4.2, as well as kinases, Hb and glycolytic enzymes (glyceraldehyde-3-phosphate dehydrogenase, phosphofructokinase, aldolase) [50,51,52]. The interaction between band 3 and glycolytic enzymes is modulated by Hb oxygenation levels and the phosphorylation status of band 3 [53]. Under hypoxic conditions, Hb binding to band 3 causes the release of glycolytic enzymes into the cytosol, thereby promoting glycolysis and 2,3-biphosphoglycerate synthesis via the Rapoport–Luebering shunt [54,55]. Conversely, Hb oxygenation is associated with a reduction in glycolytic flux, favoring the pentose phosphate pathway and subsequent NADPH production, which serves as a cofactor for various antioxidant systems [56]. The dynamic and multifunctional nature of band 3 underlies its essential role in RBC physiology, integrating transport, structural integrity, and metabolic regulation. While band 3 represents the most abundant integral protein, several other proteins contribute fundamentally to the mechanical stability, deformability, ion balance, and metabolic regulation of RBCs, often through interactions with the membrane-associated cytoskeleton. Among integral membrane proteins, glycophorins, particularly glycophorin A (GPA), are prominent sialoglycoproteins. GPA is the second most abundant transmembrane protein and plays critical roles in maintaining membrane negative charge and mediating cell–cell and cell–pathogen interactions [57]. GPA contains a heavily glycosylated extracellular domain that confers electrostatic repulsion, reducing aggregation and promoting circulatory stability. It also interacts laterally with band 3, forming the Wright blood group antigen complex and modulating anion exchange activity. Glycophorin C (GPC) and Glycophorin D (GPD) are essential for anchoring the protein 4.1R–actin–spectrin complex through interaction with protein 4.1R. This complex stabilizes the membrane in areas where band 3–ankyrin anchorage is sparse, contributing to membrane mechanical resilience and elasticity. This topic will be discussed in detail in the next section, along with peripheral proteins that form the cytoskeleton. Aquaporin-1 is another key integral protein, forming water-selective pores that regulate osmotic balance. Its function is essential for rapid water exchange during cell volume fluctuations, particularly under osmotic stress or dehydration [58].

#### 2.1.3. Physiological Functions of RBC Membrane

As a whole, the RBC membrane exemplifies the principle that form follows function. Its distinctive molecular architecture fulfills its unique necessity of mechanical flexibility, enabling RBC to recover its peculiar biconcave shape after undergoing significant deformations. Each element (i.e., lipid organization, protein composition, cytoskeletal anchorage) contributes to a membrane system that is simultaneously robust and responsive, simple in function yet complex in design. The RBC membrane performs a wide array of essential physiological functions that extend well beyond its role as a structural boundary. These functions reflect a finely tuned interplay among its lipid, protein, and cytoskeletal components. The most immediate and evident role is mechanical: due to the specialized organization of the cytoskeleton, RBC adopts a biconcave shape that maximizes the surface-to-volume ratio, enhancing gas exchange, while simultaneously maintaining exceptional deformability [59,60]. This deformability enables the cell to traverse capillaries with diameters smaller than the cell itself. Notably, this mechanical flexibility is not a purely passive property; rather, it is dynamically regulated through reversible modifications of the interactions between the lipid bilayer and the cytoskeleton, involving regulatory enzymes and intracellular ionic signals. In particular, mechanical stress activates the mechanosensitive channel PIEZO1, which allows Ca^2+^ influx. The subsequent rise in intracellular Ca^2+^ activates the Ca^2+^-dependent potassium channel KCNN4 (also known as the Gardos channel), promoting the efflux of K^+^ and water, ultimately resulting in cell shrinkage. This volume reduction protects RBC from osmotic damage [61]. A second key functional domain of the RBC membrane is its involvement in respiratory gas transport. Although Hb is the molecule directly responsible for binding O_2_ and CO_2_, the membrane actively contributes to CO_2_ transport via band 3 in response to local pCO_2_ gradients in tissues and lungs [50]. The RBC membrane also plays a fundamental role in maintaining ionic homeostasis, which is critical for cell viability. The Na^+^/K^+^-ATPase, located on the plasma membrane, sustains the electrochemical gradients of sodium and potassium, thereby preserving osmotic balance and cell volume [62]. In addition to its biophysical and transport-related functions, the RBC membrane plays a crucial role in modulating interactions with the immune system. A key protein in this context is CD47, which acts as a ligand for the SIRPα receptor on macrophages, delivering an inhibitory signal that prevents phagocytosis and thereby protects RBCs from premature removal. During senescence or under oxidative stress conditions, the surface amount or conformation of CD47 may undergo alterations, contributing to the RBC recognition as “non-self” and promoting its selective phagocytosis by macrophages. The membrane also hosts a functional antioxidant defense system. Enzymes such as CAT, GpX, and SOD may bind the inner membrane surface, thus safeguarding membrane lipids and proteins from peroxidation and oxidative denaturation [63]. Finally, the RBC membrane serves as a molecular interface between the cell and the plasma environment, influencing susceptibility to infections. Numerous pathogens, most notably *Plasmodium falciparum*, selectively bind to membrane proteins such as GPA to enter the host cell, exploiting sialylated glycoproteins as receptors [45]. In this respect, the RBC membrane functions not only as a structural barrier but also as a critical site of molecular vulnerability. The delicate and complex arrangement of RBC membranes is highly susceptible to disruption by oxidative stress (Table 1). In the following sections, we will examine in detail how oxidative stress drives these disturbances, focusing on lipid peroxidation and oxidation of membrane proteins.

### 2.2. Lipid Peroxidation

Lipid peroxidation is a critical process through which oxidative stress compromises the structural and functional integrity of RBC membranes. It predominantly affects PUFAs, long-chain fatty acids with multiple double bonds such as linoleic, arachidonic, and docosahexaenoic acids, rendering the membrane susceptible to oxidative damage [75]. The peroxidation cascade begins when ROS (notably OH^•^) removes a hydrogen atom from the methylene group of a PUFA (typically arachidonic or docosahexaenoic acid), thus producing a lipid radical (L^•^). This radical rapidly reacts with O_2_ to form a lipid peroxyl radical (ROO^•^), which propagates the reaction by abstracting hydrogen from adjacent PUFAs, generating new lipid peroxides (ROOH) and radicals. This sequence of events characterizes the propagation phase of lipid peroxidation and is considered a hallmark of ROS-induced membrane injury. Lipid hydroperoxides undergo further iron-catalyzed decomposition into highly reactive aldehydes and electrophilic compounds such as malondialdehyde (MDA) and 4-hydroxynonenal (4-HNE) [76]. Malondialdehyde, for instance, can covalently bind to membrane proteins and phospholipids, leading to crosslinking and Schiff base formation with lysine residues [77,78]. Similarly, 4-HNE preferentially reacts with cysteine, histidine, and lysine residues in proteins, altering their structure and function [79,80]. These oxidative modifications impair lipid–lipid interactions, compromise membrane protein function and cytoskeletal anchorage, and ultimately lead to alterations in membrane fluidity, thickness, permeability, and ionic homeostasis [81,82]. Moreover, oxidative stress inhibits flippase activity, resulting in PS externalization that, along with the loss of GPA and reduction in protein CD47 expression, has been observed in senescent RBCs, concurrently with increased ROS accumulation [83,84,85]. To mitigate these oxidative insults, RBCs rely on a robust antioxidant defense system comprising GpX, CAT, and SOD, which work synergistically to neutralize ROS and reduce lipid hydroperoxide levels [10]. However, once peroxidation occurs, damaged phospholipids must be repaired or replaced to restore membrane functionality. This is achieved through the Lands cycle [86], a phospholipid remodeling pathway that involves two key steps: (1) deacylation of oxidized phospholipids by phospholipase A_2_, yielding a lysophospholipid and a free oxidized fatty acid; and (2) reacylation of the lysophospholipid by lysophospholipid acyltransferases, typically incorporating a non-oxidized PUFA such as arachidonic or docosahexaenoic acid [86,87]. This process, originally described by William E.M. Lands, maintains bilayer integrity under oxidative stress. Despite the lack of nuclei and mitochondria, mature RBCs retain sufficient cytosolic enzymatic activity for partial operation of the Lands cycle, depending on the availability of ATP, coenzyme A, and acyl-CoA derivatives [56]. Impairment of this repair mechanism, either due to ATP depletion, GSH exhaustion, or inherited genetic defects in antioxidant enzymes, leads to the accumulation of oxidized lipids and membrane vesiculation [56,75].

### 2.3. Protein Oxidation

Oxidative protein post-translational modifications represent one of the primary mechanisms by which oxidative stress impairs RBC functionality and lifespan [88]. Membrane proteins are susceptible to both direct and indirect oxidation elicited by ROS, including O_2_^•−^, H_2_O_2_, and OH^•^ [89]. Major oxidative modifications include disulfide bond formation and thiol oxidation, carbonylation, tyrosine nitration and dityrosine cross-linking, as well as glycation and the formation of advanced oxidation protein products (AOPPs). Cysteine residues in integral membrane proteins, such as band 3, Na^+^/K^+^-ATPase pump, CD47, and aquaporin-1, are particularly susceptible to ROS, which can initiate thiol oxidation either through direct radical-mediated attack or via reaction with secondary oxidants such as H_2_O_2_ or lipid hydroperoxides [26,32,88]. Initial oxidation yields sulfenic acid intermediates (–SOH), which may reversibly react with neighboring thiols to form intra- or intermolecular disulfide bonds [90]. This redox-induced cross-linking alters the tertiary and quaternary structure of proteins, often leading to clustering, misfolding, or aberrant interactions with other membrane components [91]. When oxidative stress exceeds the antioxidant buffering capacity of the cell, thiol groups can undergo further irreversible oxidation to sulfinic (–SO_2_H) and sulfonic (–SO_3_H) acids [92,93]. These end-stage modifications result in permanent structural alterations and loss of protein function, effectively tagging the membrane protein for proteolytic degradation or vesiculation [94,95]. Protein carbonylation represents a non-enzymatic modification and involves the covalent introduction of carbonyl groups (–C=O), typically aldehydes or ketones, into amino acid side chains (primarily proline, arginine, lysine, and threonine) through two major pathways: direct oxidation by ROS and secondary modification via lipid peroxidation-derived aldehydes MDA and 4-HNE [89]. In RBC membranes, carbonylation preferentially targets high-abundance integral proteins such as band 3, CD47, and GPA, which are chronically exposed to oxidative insult due to the high O_2_ tension and heme-derived ROS. Carbonyl groups can also form Schiff bases and Michael adducts with nucleophilic residues of neighboring proteins or phospholipids, promoting protein cross-linking and aggregation. Unlike reversible oxidative modifications such as disulfide bond formation or sulfenylation, carbonylation is an irreversible process that serves as a marker of cellular aging and oxidative stress [96]. Nonetheless, once carbonylation has occurred, the only physiological route of resolution is cell removal via the reticuloendothelial system [97,98]. Elevated carbonyl content has been correlated with increased RBC vesiculation and clearance in diseases such as diabetes mellitus, β-thalassemia, and sepsis, where oxidative stress is a major pathogenic component [99,100]. Tyrosine nitration involves the addition of a nitro group (–NO_2_) to the ortho position of the aromatic ring of tyrosine residues, forming 3-nitrotyrosine (3-NT) [101]. This reaction is predominantly mediated by peroxynitrite (ONOO^−^), a potent reactive nitrogen species formed by the diffusion-limited reaction of O_2_^•−^ with nitric oxide (NO) [102]. In RBC membranes, tyrosine nitration has been observed in key transmembrane proteins such as band 3 and CD47, especially under pathological conditions characterized by inflammation, hypoxia, or oxidative imbalance, including sepsis, diabetes, and hemoglobinopathies [103,104]. Nitration can lead to steric hindrance and alteration of protein conformation, disrupting binding interfaces or enzyme active sites and often impairing protein-protein interactions and signal transduction [105]. Moreover, tyrosine nitration is irreversible under physiological conditions and serves as a molecular footprint of oxidative damage, often associated with enhanced immunogenicity and increased macrophage recognition of altered RBCs [96]. Dityrosine cross-linking refers to the formation of covalent bonds between two tyrosine residues on the same or different proteins, catalyzed by oxidative conditions, particularly through the action of peroxidases or ONOO^−^ [106,107]. Both nitration and dityrosine formation are enhanced in disease states marked by iron overload, increased NO production, or chronic inflammation, as observed in β-thalassemia and sickle cell disease [108]. Glycation is a spontaneous, non-enzymatic reaction between reducing sugars (mainly glucose) and the free amino groups of proteins, lipids, or nucleic acids [109]. In RBCs, the reaction begins with the formation of a Schiff base, which rearranges to a more stable Amadori product, namely glycated Hb (HbA1c). Over time, these intermediates undergo oxidation, dehydration, and rearrangement to form advanced glycation end-products (AGEs) [110]. AGEs promote cross-linking of membrane proteins and the generation of ROS via binding to their receptor (RAGE), thus creating a feed-forward loop of oxidative stress [111]. In RBC membranes, glycation primarily affects band 3 and extracellular domains of glycophorins [112]. Advanced oxidation protein products (AOPPs) are dityrosine-containing, cross-linked protein aggregates formed primarily through the action of chlorinated oxidants such as hypochlorous acid and myeloperoxidase-derived oxidants, which modify plasma or membrane proteins during oxidative stress [113]. In RBC membranes, AOPPs are generated from oxidized albumin, globin chains, and structural proteins like band 3, leading to protein aggregation, transporter and surface receptor loss of function, and altered membrane fluidity [114]. Advanced oxidation protein products act as both markers and mediators of oxidative stress. They stimulate NADPH oxidase activity and amplify ROS production [115]. Collectively, these oxidative changes result in structural and functional alterations of integral membrane proteins, i.e., membrane transport systems. In particular, such structural perturbations have direct consequences on band 3 anion exchange capability, leading to reduced efficiency in Cl^−^/HCO_3_^−^ transport across the RBC membrane, essential to CO_2_ transport and acid–base homeostasis [21,51]. Moreover, thiol oxidation of band 3 is associated with the formation of high-molecular-weight aggregates [116], which not only impairs anion exchange capability but also renders band 3 a neoantigen promoting opsonization and splenic macrophage erythrophagocytosis [117]. The function of Na^+^/K^+^-ATPase is closely tied to cytoplasmic rheology and the surface area-to-volume ratio, factors that critically influence deformability [118]. Under oxidative stress conditions, Na^+^/K^+^-ATPase activity is generally diminished. Several oxidative modifications, such as carbonylation and nitration of amino acid residues, disrupt the structural integrity and catalytic function of the pump [119]. Additionally, oxidative stress leads to ATP depletion, limiting the energy substrate required for pump activity [120]. Together, these factors reduce ion transport efficiency, resulting in the intracellular accumulation of Na^+^, loss of K^+^, and subsequent osmotic imbalance [121]. Interestingly, free intracellular iron exhibits a paradoxical effect by selectively enhancing Na^+^/K^+^-ATPase activity through the oxidation of critical enzyme thiol groups. This oxidative modification appears to transiently stimulate pump function, possibly as a compensatory mechanism to counterbalance ionic dysregulation [71,72]. However, iron redox activity also predisposes the cell to further oxidative damage, reflecting a complex regulatory balance in which iron exerts dual roles: facilitating pump activation while promoting oxidative stress [50]. Oxidative modification of CD47 alters its extracellular conformation and disrupts its interaction with SIRPα on macrophages, thereby compromising the “do not eat me” signal [122]. Similarly, oxidation of aquaporin-1 impairs its water channel conductance, contributing to osmotic dysregulation under oxidative stress [74]. Beyond functional impairments in individual proteins, oxidative modifications affect the biophysical properties of the whole membrane [82]. Disulfide bond formation and protein carbonylation alter lipid–protein and membrane–cytoskeleton interactions, increasing membrane rigidity and reducing lateral mobility [68,123]. These changes compromise cellular deformability and hinder microvascular transit [123].

## 3. Oxidative Stress at Cytoskeleton Level: Molecular Targets and Underlying Mechanisms

### 3.1. Structure and Functions of the RBC Cytoskeleton

The biophysical properties and spatial distribution of cytoskeletal proteins are fundamental to maintaining the characteristic RBC biconcave shape, essential to their transit through narrow microvascular networks and to the preservation of membrane integrity in the systemic circulation, thus reducing early splenic clearance [67]. Structurally, the RBC cytoskeleton consists of α and β spectrin tetramers, which assemble into a pseudo-hexagonal lattice anchored to the plasma membrane by two main complexes (Figure 2): the ankyrin and the junctional complex, both stabilized by accessory proteins such as adducin, tropomodulin, and dematin, which form interconnected tethers approximately 80 nm apart [38,124].

Within the first complex, ankyrin binds the cytosolic domain of dimeric band 3 via its AR 17–20 repeats [125]. Concurrently, the central ZU5 ankyrin domain engages β-spectrin (repeats 14–15) with high affinity (~15 nM), thus establishing a robust support for the band 3–ankyrin–spectin connection [126]. This ternary complex is further stabilized by protein 4.2, which directly interacts with both band 3 (residues 200–211) and ankyrin (residues 187–200) [127]. The junctional complex consists of short actin filaments (5–7 monomers) that act as nodal points for α- and β-spectrin tetramer attachment [128]. This interaction is ensured by adducin, which promotes actin polymerization and stabilizes its interaction with spectrin and tropomodulin, regulating the length of the actin filament by capping the minus terminal [129]. The link between band 3 and the junctional complex is achieved through a series of indirect protein-protein interactions that mediate membrane–cytoskeleton linkage at actin–spectrin nodes. Specifically, band 3 interacts with protein 4.1R through a molecular platform composed of dematin, adducin, and additional proteins such as p55 and GPC [130]. GPC directly interacts with protein 4.1R, which in turn stabilizes the bond between actin and spectrin [130]. While band 3 does not directly interact with 4.1R, it participates through lateral associations with GPC to create membrane microdomains of protein aggregates that facilitate junctional assembly. Moreover, p55 binds to both protein 4.1R and GPC, as well as the band 3 cytosolic domain, reinforcing the membrane–cytoskeleton scaffold at actin–spectrin nodes [131]. This intricate architecture enables cytoskeletal anchorage to the membrane in both vertical (via ankyrin) and lateral (via junctional complexes) orientations, thereby distributing mechanical forces and contributing to the biomechanical resilience of RBCs during their transit through capillaries. Therefore, while the ankyrin complex ensures vertical mechanical stability between spectrin and the membrane, the junctional complex provides a planar network that is securely anchored [132,133]. In addition to its association with the ankyrin and junctional complexes, approximately 27% of band 3 is unanchored to the cytoskeleton and therefore free to diffuse laterally in the membrane [134,135].

This arrangement is critical for preserving the structure and consequently the functionality of RBCs [125]. However, chronic oxidative stress levels may elicit redox-sensitive molecular responses that could compromise cytoskeletal components, leading to impaired deformability of RBCs [82].

### 3.2. Oxidative Alteration of the RBC Cytoskeleton

The mechanisms underlying oxidative modifications of cytoskeletal proteins include the formation of disulfide bonds, carbonylation, nitrification, loss of ubiquitination, and glutathionylation. The oxidation of thiol groups in cysteines results in the formation of intra- and intermolecular disulfide bonds. For instance, H_2_O_2_ promotes the formation of intermolecular disulfide bonds between Cys374 actin residues, leading to monomer interconnections, reduced polymerization, and aggregate accumulation [136]. Spectrin also contains cysteines prone to reactive species-induced disulfide bond formation, which leads to cross-linking, aggregation, and a significant loss of methionine and histidine. This process decreases RBC deformability by around 10% [137]. Carbonylation clearance causes an aberrant reorganization of the F-actin network, the inactivation of the polymerizing function, and the formation of non-functional aggregates, associated with irreversible cellular dysfunction [138]. Furthermore, carbonylation of critical components such as band 3, protein 4.1R, ankyrin, and spectrin, often triggered by toxic aldehydes (e.g., 4-HNE), reduces membrane adhesion to the cytoskeleton, thus increasing fragility and predisposing RBCs to hemolysis [69]. Nitrosative modifications through nitrification compromise the conformation and binding capability of cytoskeletal proteins [139]. The main targets of this reaction are actin and spectrin. As concerns actin, the formation of 3-NT leads to depolymerization and, consequently, to the destabilization of F-actin filaments, thereby impairing the integrity of the cytoskeletal network [140]. Similarly, the structural organization of spectrin is irreversibly disrupted in this scenario, inducing mechanical instability and structural reorganization. Ubiquitination is a post-translational modification essential to the maintenance of the proper dynamic cytoskeleton structure [141]. Unlike the classic ubiquitin–proteasome system associated with protein degradation, the ubiquitination process in RBCs predominantly acts as a regulatory mechanism that modulates the interactions between cytoskeletal proteins and membranes [142]. In particular, α-spectrin displays intrinsic E2/E3 ubiquitin ligase, able to catalyze its own ubiquitination and that of other partner proteins, such as ankyrin [142]. This activity is driven by specific cysteine residues, especially Cys2071 and Cys2100, whose integrity is crucial for the ubiquitinating function of the protein. Once ubiquitinated, the ability of spectrin to bind to other components of the cytoskeletal complex is modified, and the ordered dissociation of key nodes such as spectrin–adducin–actin is promoted, thereby facilitating a dynamic network turnover [143]. The ubiquitination pattern declines over time: during RBC aging, the amount of ubiquitinated spectrin progressively decreases, with a reduction of about 50% in senescent cells. This reduction correlates with a decrease in E2/E3 activity, suggesting that ubiquitination contributes to the maintenance of mechanical plasticity in the early phase of RBC lifespan. Conversely, the loss of spectrin via ubiquitination may predispose older cells to splenic removal in order to eliminate damaged and non-functional cells [144]. In pathological conditions associated with chronic oxidative stress, such as sickle cell anemia [143], this fragile balance is severely compromised, as in the case of a redox imbalance promoting glutathionation (a covalent binding between GSH and proteins via a thioether bond) of reactive cysteine residues of spectrin, thus disrupting its E2/E3 function and preventing both auto-ubiquitination and the modification of other cytoskeletal partners, particularly ankyrin [144]. The main outcome is an altered architecture of the sub-membranous network, with accumulation of stable and dissociation-resistant complexes and loss of controlled remodeling and of the elastic response to hemodynamic stresses [145,146]. Collectively, these oxidative changes result in impaired rheological properties, increased osmotic fragility, exposure of neoepitopes that promote macrophage recognition, and, subsequently, activation of early removal pathways [147]. Furthermore, alterations in the lateral distribution of membrane proteins following cytoskeletal damage promote the formation of micro-vesicles enriched in Hb and PS, which further propagate systemic pro-inflammatory and procoagulant responses [38]. These structural alterations cause abnormal RBC shapes, as summarized in Table 2.

Pathologically, these changes contribute to several hematological and systemic disorders. In sickle cell anemia, sustained oxidative stress results in an increased nitrification of actin and spectrin, associated with cell rigidity, vascular adhesiveness, and occlusion [156,157]. In metabolic diseases such as diabetes mellitus, ROS overproduction promotes carbonylation and cross-linking of cytoskeletal proteins, with similar effects on RBC deformability and blood viscosity [158]. In enzyme deficiencies such as glucose-6-phosphate dehydrogenase (G6PD) deficiency, the failure to regenerate NADPH exposes the cytoskeleton to irreversible oxidative injury, with the formation of Heinz bodies and massive splenic sequestration [159]. Based on these findings, RBC cytoskeleton should not be considered a passive structure scaffold but rather as a dynamic sensor and effector of the cellular redox homeostasis. Oxidative changes modulate cytoskeleton–membrane interactions, membrane transport protein distribution, and intracellular signaling, ultimately influencing RBC fate. Furthermore, the effect of antioxidant compounds on band 3 appears critical, as its interaction with spectrin and ankyrin is essential for maintaining the rheological properties of RBCs [57]. This protective activity may hold significant therapeutic potential for the treatment and prevention of chronic oxidative stress-associated conditions and cellular dysfunction, especially in highly susceptible cells such as RBCs.

## 4. Functional Role of Cytosolic Components in RBC Response to Oxidative Stress

The cytosolic component of RBCs hosts antioxidant enzymes such as SOD and CAT, which are essential to cell protection against oxidative stress, as well as various kinases involved in energy metabolism regulation and the preservation of cellular integrity. These components work synergistically to neutralize ROS, thereby preserving the structural and functional integrity of the RBC membrane and intracellular proteins [160]. The cytosolic content of RBCs also comprises hemoglobin, which can act both as a source and a scavenger of ROS, while antioxidant enzymes catalyze crucial detoxification reactions [10,161,162]. Protein kinases, such as PKC, are involved in regulating the oxidative stress response by modulating the phosphorylation state of target proteins implicated in cytoprotection [163,164]. The antioxidant response of RBC is not static but varies depending on the nature and magnitude of the oxidative stress. Conditions such as pH fluctuations, elevated glucose levels, cellular aging, chronic inflammation, and systemic diseases significantly influence the activity of these antioxidant defenses [165,166]. For instance, in hyperglycemic conditions typical of diabetes mellitus, increased ROS production is accompanied by an impaired antioxidant enzyme activity, contributing to RBC fragility and systemic oxidative damage [167,168]. Similarly, aging and chronic inflammatory diseases are associated with a progressive decline in RBC antioxidant capacity due to post-translational modifications of cytosolic proteins and increased susceptibility to oxidative damage [169]. Given the dynamic nature of RBC antioxidant responses to various oxidative challenges, as previously discussed, multiple studies have sought to further elucidate the underlying molecular mechanisms. Accordingly, the following section will provide an in-depth analysis of the functional roles of key cytosolic proteins and enzymes, with the objective of highlighting the remarkable adaptive efficiency of these uniquely specialized cells in responding to oxidative stress under physiological and pathological conditions.

### 4.1. Role of Hb in Redox Homeostasis

Hemoglobin, the most abundant cytosolic protein in RBC, plays a dual role in oxidative balance. On one hand, it can generate ROS through auto-oxidation processes, particularly during O_2_ transport and deoxygenation [26]. On the other hand, it serves as a redox-active molecule capable of scavenging ROS and participating in electron transfer reactions. However, under pathological or stressful conditions, oxidative modifications of Hb through metHb or hemichrome formation not only impair its O_2_-carrying capacity but also trigger membrane damage, protein aggregation, and RBC senescence [170]. These oxidative forms of Hb tend to associate with band 3, altering its function and stability through direct interactions with the cytoskeleton. Supporting this, Ivanov et al. demonstrated that under acidic conditions, Hb becomes oxidized and generates ROS capable of damaging RBC membranes [171]. Additionally, Morabito et al. showed that during oxidative stress, the oxidation of thiol groups in membrane proteins, particularly in band 3, disrupts anion exchange, promoting the formation of oxidative clusters with denatured Hb and resulting in increased membrane rigidity [172]. To discern whether oxidative damage primarily affects the membrane or the cytosolic compartment, these authors [172] have used Hb-free resealed ghosts, artificial RBC membranes devoid of cytoplasmic components. Notably, these ghosts did not exhibit impaired anion exchange after exposure to 300 μM H_2_O_2_, suggesting that the oxidative damage is mediated mainly by intracellular components, particularly Hb. The data support a mechanism according to which H_2_O_2_ crosses the RBC membrane and reacts with cytosolic Hb, generating ROS that secondarily damage membrane proteins such as spectrin and band 3 [172].

Additional insights come from preconditioning studies, whereby RBCs subjected to sublethal oxidative stimuli develop enhanced resistance to subsequent stressors. In this context, H_2_O_2_ at millimolar concentrations primarily targets Hb before triggering hemolysis. However, within the range of 10–300 μM H_2_O_2_, no significant alteration in Hb or metHb levels was detected, even in the presence of CAT inhibitors like NaN_3_ or 3-AT [21]. This suggests that moderate H_2_O_2_ exposure does not necessarily induce Hb oxidation, although subtle interactions cannot be completely excluded. In chronic exposure to D-Galactose (D-Gal), Hb emerges not only as a primary molecular target of redox imbalance but also as a central mediator of RBC structural and functional decline. While short-term D-Gal treatment does not significantly alter the redox state of cytosolic Hb, prolonged exposure (24 h) initiates a cascade of changes beginning with non-oxidative glycation [173]. This modification affects Hb conformation and reactivity, potentially disturbing its high-affinity interaction with a membrane transporter crucial for anion exchange, such as band 3. Remarkably, this glycation occurs without any detectable increase in metHb, indicating that oxidative conversion of Hb is not a prerequisite in the early stages of D-Gal-induced damage. Instead, the high intracellular reactivity and rapid uptake of D-Gal likely explain the accelerated onset of glycation. These early molecular alterations in Hb preceded more overt oxidative events and suggest that Hb plays a proactive role in sensing and propagating metabolic stress within RBC. This interpretation is supported by data showing that after 24 h of incubation with D-Gal at concentrations of 25, 35, 50, and 100 mM, no significant elevation in metHb levels was observed. However, a marked increase in HbA1c was detected, signifying that early-stage glycation is underway, even in the absence of direct oxidative damage. Thus, the aging-mimetic effect of D-Gal seems to follow a stepwise trajectory, beginning with Hb glycation, followed by disruption of its interaction with band 3, and culminating in oxidative damage at the membrane level. Hemoglobin, therefore, stands at the intersection of oxidative stress, protein glycation, and membrane destabilization, orchestrating key events in RBC senescence. This role is further underscored by evidence linking environmental stressors, such as heavy metal exposure, to RBC dysfunction. Notably, mercury, whose human exposure has dramatically increased, preferentially accumulates in RBCs, inducing morphological changes and enhancing their procoagulant activity. Recent findings by Perrone et al. demonstrate that exposure to mercury significantly elevates oxidative stress in RBCs through increased production of ROS, which in turn promotes metHb formation [174]. This oxidized form of Hb is incapable of binding oxygen, thereby compromising RBC functionality and exacerbating cellular aging [175]. However, Hb slow auto-oxidation represents a significant endogenous source of oxidative stress, leading to the formation of metHb and ROS [176]. Under acute oxidative stress conditions such as those experimentally induced by AAPH, a widely used pro-oxidant agent to model free radical overproduction, metHb levels significantly rise, contributing to membrane structural alterations and functional impairment in RBCs. Notably, metHb binds to the N-terminal cytoplasmic domain of the band 3, promoting its clustering [69]. This aggregation affects membrane architecture and may facilitate the recognition and premature clearance of senescent RBCs from the circulation [69]. Beyond its structural role, Hb is involved in the O_2_-dependent regulation of glycolysis by competing with glycolytic enzymes for binding sites on band 3. In oxidative conditions, these interactions are disrupted, altering RBC energy metabolism and contributing to cellular dysfunction [177]. The relationship between increased oxidative stress and various pathological conditions is an area of continuous research and growing clinical interest. In particular, several studies have explored the link between oxidative stress and the altered rheological properties of RBCs in individuals with β-thalassemia minor (β-Thal^+^), although this association remains only partially understood. During the clinical course of β-Thal^+^, elevated levels of ROS can compromise RBC deformability through oxidative processes such as protein degradation, metHb formation, and subsequent hemolysis. The degradation of unstable Hb, along with iron overload, represents a major source of oxidative injury in RBCs of β-Thal^+^ subjects. Under physiological conditions, human RBCs contain approximately 3% metHb, a form typically reduced back to functional Hb by NADH-dependent cytochrome b5 reductase. However, in β-Thal^+^ cells, excessive ROS levels likely promote the oxidation of Hb to metHb, thereby heightening RBC vulnerability to oxidative damage and impairing their physiological functions. Supporting this, Spinelli et al. demonstrated that Hb oxidation is associated with a reorganization of band 3 protein into clusters, likely through dimer or oligomer formation, without a reduction in overall band 3 expression levels [72]. This interplay between oxidative stress and RBC dysfunction is not limited to hemoglobinopathies such as β-Thal^+^ but may also contribute to RBC impairment in metabolic disorders like prediabetes and type 2 diabetes [178,179]. Before developing type 2 diabetes, individuals often pass through a prediabetic stage characterized by elevated blood glucose levels in the absence of overt symptoms. Impaired glucose tolerance and impaired fasting glucose are two key markers of this condition, which already carries a significantly increased risk of progression to type 2 diabetes, as well as cardiovascular and cerebrovascular complications [180]. Although the precise mechanisms underlying these risks are not fully elucidated, hyperglycemia-induced oxidative stress is considered a major contributing factor [181]. Excessive ROS production in this context can oxidize Hb into metHb, rendering RBCs more susceptible to oxidative damage [182]. Moreover, Hb oxidation facilitates the formation of hemichromes, which serve as precursors to Heinz body aggregates that compromise the structural integrity and oxygen-carrying capacity of RBCs (Figure 3) [183]. In diabetic patients, the accumulation of these Hb degradation products can severely impair tissue oxygenation, further exacerbating disease complications [181].

### 4.2. Endogenous Antioxidant System and Its Regulation

The RBC cytosol contains a highly specialized and tightly regulated antioxidant defense system, composed of key enzymes such as SOD1, CAT, GPx, and GSH. These components synergistically act to neutralize ROS, including O_2_^•−^, H_2_O_2_, and lipid peroxides, thereby preserving redox homeostasis [10]. Glucose-6-phosphate dehydrogenase also an essential role in this system by sustaining NADPH production, a critical cofactor for GSH regeneration and other reductive reactions [3]. Disruptions in this finely tuned antioxidant network are often induced by pathological stressors such as hyperglycemia, acidosis, or systemic inflammation, compromising RBC redox balance and finally leading to increased vulnerability to hemolysis and protein oxidation [179,185]. Among cytosolic antioxidant enzymes, CAT plays a key role in neutralizing H_2_O_2_ by rapidly converting it into H_2_O and O_2_, thereby preventing the formation of harmful free radicals. Its functional significance is underscored by the evidence that CAT inhibition exacerbates oxidative damage to the RBC membrane, particularly compromising band 3 activity. Interestingly, exposure to low-to-moderate concentrations of H_2_O_2_ (10–300 μM) elicits a measurable increase in CAT activity, reflecting a swift functional upregulation of the enzyme and reinforcing its role in the adaptive cellular response to oxidative stress [172]. This response forms the basis of a phenomenon of preconditioning. Although enzymes such as SOD and GPx may also contribute to this protective mechanism, CAT emerges as the principal mediator of this adaptation. To reinforce the pivotal role of cytosolic enzymes in maintaining redox homeostasis, studies performed on experimental models of diseases such as leishmaniasis have shown that oxidative stress in RBCs does not necessarily lead to GSH depletion. Instead, a compensatory increase in antioxidant enzymes like CAT and Prx is often observed, indicating an alternative defense mechanism that supports RBC survival under inflammatory and oxidative stress conditions [186]. Despite these compensatory mechanisms, the maintenance of intracellular GSH levels remains a critical determinant of RBC redox homeostasis. In this context, the exposure of RBCs to oxidants such as N-ethylmaleimide (NEM) results in rapid depletion of GSH, thereby weakening the cell capacity to neutralize ROS. This vulnerability is particularly evident in RBCs from diabetic patients, which display a significantly reduced GSH/GSSG ratio, reflecting an impaired redox buffering system [187]. These cytosolic redox disturbances precede any sign of lipid peroxidation and contribute to functional alterations in the membrane, including increased band 3-mediated anion exchange. Similar redox imbalances have been reproduced in vitro by exposing healthy RBCs to high glucose concentrations (15–35 mM), reinforcing the idea that oxidative stress originates in the cytosol and only later affects membrane structure and function [188].

Inflammatory processes also exert a substantial oxidative impact on RBCs. Acute inflammation has been shown to decrease intracellular GSH levels, further compromising cytosolic redox homeostasis [189]. The impact of acute inflammation, revealed by C-reactive protein (CRP) plasma levels, has been studied also on RBCs. Under acute inflammation, a significant decreased GSH content was observed [190]. These findings align with evidence from chronic oxidative stress models, such as D-Gal administration. In RBCs exposed to D-Gal, SOD activity increases O_2_^•−^ production, which is then converted to H_2_O_2_. Under physiological conditions, CAT mitigates this stress, but the inhibition of CAT with 3-AT significantly amplifies the oxidative burden. This leads to a reduction in band 3 function, reinforcing the role of cytosolic enzymatic defenses in preventing oxidative damage and maintaining RBC functionality. After treatment with D-Gal, a notable decrease in the GSH/GSSG ratio was also observed, indicating that oxidative imbalance was effectively induced and that the RBC antioxidant defense has been compromised [173]. This reduction in the GSH/GSSG ratio not only reflects a decline in cellular redox buffering capacity but also demonstrates that high concentrations of D-Gal directly impact both the RBC membrane and key intracellular components, including the antioxidant system [173]. Importantly, this finding is in line with data from in vivo studies reporting a significant age-related decline in GSH levels in the brain, accompanied by increased oxidation to GSSG and a reduction in the GSH/GSSG ratio [191]. During aging, the progressive depletion of GSH not only weakens redox homeostasis but also amplifies the vulnerability of Hb to oxidative damage and functional decline [192]. A significant decrease in the GSH/GSSG ratio was observed in erythrocytes exposed to 10 μM HgCl_2_, confirming the detrimental effect of oxidative processes on the antioxidant system [174]. Antioxidant defense in RBCs critically relies on both enzymatic and non-enzymatic systems. Among the key enzymatic components, SOD and CAT play central roles in neutralizing ROS and maintaining redox homeostasis. Under pro-oxidant conditions, such as exposure to high concentrations of D-Gal, SOD and CAT activities increase in an attempt to scavenge excessive free radicals [193,194]. However, enzyme upregulation is insufficient to counterbalance oxidative damage, as indicated by increased lipid peroxidation and protein carbonylation levels [67]. This paradoxical scenario reflects a state of antioxidant system exhaustion, where sustained enzyme activity is unable to prevent membrane injury and functional decline. As previously mentioned, Remigante et al. utilized AAPH to mimic a potent oxidative stress condition, contributing to valuable insights into oxidative stress-related pathophysiological mechanisms. Exposure of RBCs to AAPH resulted in a significant reduction (~45%) of intracellular GSH levels [148]. In a different study on RBCs from prediabetic subjects, the enzymatic activities of SOD and CAT were significantly increased compared to healthy controls [195], putatively accounting for a compensatory response of the endogenous antioxidant system to counteract elevated ROS levels. However, this defense appears to be insufficient, as GSH depletion is closely related to increased metHb. Indeed, a significant reduction in the GSH/GSSG ratio was observed in prediabetic RBCs, reflecting an altered redox balance [195]. These findings are consistent with previous evidence in diabetic patients, where similar alterations in the GSH/GSSG system have been reported [196].

### 4.3. Protein Kinases and Phosphorylation-Dependent Signaling Pathways

Despite lacking nuclei, RBCs retain a surprisingly dynamic repertoire of cytosolic signaling mechanisms mediated by reversible protein phosphorylation. This post-translational modification serves as a ubiquitous regulatory tool in cellular physiology, influencing key processes such as signal transduction, ion and metabolite transport, cytoskeletal remodeling, metabolism, and cell volume control. There are two critical sites of cytoskeletal interaction targeted by phosphorylation: the first involves the assembly of the spectrin/actin/protein 4.1 network, while the second one concerns the anchoring of this network to the membrane via the band 3/ankyrin complex [170]. The interactions between cytoskeletal and membrane proteins are key determinants of RBC membrane stability and influence the cell’s ability to deform under shear stress. These protein–protein interactions are tightly regulated by post-translational modifications, primarily through phosphorylation, which induces conformational changes in the protein structure [197]. In RBCs, the protein kinase/phosphatase network, though limited by the absence of transcriptional regulation, remains functionally active and responsive to physio-pathological stimuli [198,199]. Phosphorylation of cytoskeletal proteins such as spectrin, band 3, adducin, and protein 4.1R plays a central role in modulating RBC deformability, membrane stability, and interactions with the surrounding microenvironment [130,200]. Erythrocyte cytoskeletal proteins are phosphorylated through serine, threonine, or tyrosine residues by a number of different kinases and are dephosphorylated by various phosphatases. The balance between the activities of protein kinases and phosphatases regulates the phosphorylation state of a protein. Protein kinases previously defined in RBCs are PKC, protein kinase A (PKA), casein kinases I and II, Syk, Lyn, Hck-Fgr, and Fyn [199]. Specific kinases, including PKC, PKA, casein kinases, and MAP kinases, have been identified in mature RBCs and have been shown to translocate between the cytosol and membrane [197]. For example, PKC-mediated phosphorylation of adducin alters its affinity for spectrin and actin, thereby impacting the mechanical properties of the membrane [201].

Moreover, recent phosphoproteomic studies have revealed that aged or stressed RBCs display distinct phosphorylation signatures, suggesting a critical role for kinase activity in aging-related changes and adaptive responses to metabolic stress [202]. Notably, even in the absence of de novo protein synthesis, RBCs maintain a rich complement of kinases, phosphatases, and second messengers in both membrane and cytosolic fractions, underscoring the importance of post-translational signaling in sustaining RBC function [197]. Interestingly, Syk activation is highly sensitive to the oxidative context. In RBCs exposed to high concentrations of H_2_O_2_ (300–600 μM), Syk expression is upregulated [21,203]; however, this effect does not consistently translate to increased tyrosine phosphorylation (P-Tyr), suggesting that kinase presence may not necessarily correspond to a functional activation. Possible explanations include (1) mild oxidative conditions destabilizing proteins without triggering phosphorylation; (2) oxidative modifications hindering Syk interaction with band 3 [116]; or (3) phosphorylation occurring on non-tyrosine residues, such as serine. This scenario is different from that one related to more aggressive oxidants like ONOO^−^, which robustly stimulate P-Tyr signaling via band 3 [204]. Further evidence comes from adaptive responses such as preconditioning. Remarkably, this phenomenon in anucleated RBCs does not appear to involve tyrosine phosphorylation but is rather mediated by CAT activity [21], highlighting a unique antioxidant-centered mechanism distinct from that in nucleated cells. Studies in disease contexts support the centrality of phosphorylation signaling. In canine leishmaniasis, dogs with lower oxidative stress levels (low plasma MDA) showed reduced P-Tyr despite unchanged levels of metHb, GSH, or membrane thiol groups. The observed enhanced band 3 function suggested that P-Tyr dynamics are closely tied to RBC functionality [186]. Mechanistically, redox regulation of band 3 phosphorylation involves two key kinases: Lyn, which phosphorylates Tyr359, and Syk, which targets Tyr8 and Tyr21 [205]. These phosphorylation events interfere with cytoskeletal anchoring by disrupting spectrin/band 3 interactions, thus altering RBC shape and deformability [206]. For instance, oxidative stress induced by AAPH significantly increased band 3 tyrosine phosphorylation [148]. Finally, in β-Thal^+^ RBCs, elevated oxidative stress provokes intense band 3 phosphorylation, driven by Syk recruitment and suppression of phosphatase [72]. This hyperphosphorylation leads to structural and functional alterations: disruption of ankyrin-mediated cytoskeletal connections and impaired anion transport, both of which compromise efficient tissue oxygenation.

## 5. Multilevel Protective Roles of Natural Antioxidants in Preserving RBC Structure and Function Under Oxidative Stress

This present section aims to discuss the antioxidant properties of molecules such as polyphenols, quercetin, melatonin, anthocyanins, and vitamins C and E. These compounds, selected based on their scientific relevance, proven efficacy, and complementary mechanisms in protecting RBCs against oxidative stress (Figure 4), are among the most extensively studied natural antioxidants acting at multiple levels, from direct neutralization of ROS and nitrogen species to the preservation of membrane lipid and protein integrity, thus maintaining cytoskeletal function and redox-sensitive signaling pathways [207]. The selection is further guided by growing evidence of their efficacy in pathophysiological models, including diabetes, chronic inflammation, and premature aging characterized by impaired endogenous antioxidant defenses [208,209,210,211].

Polyphenolic compounds, widely distributed in plant-derived foods, have shown remarkable efficacy in protecting RBCs from oxidative damage through a variety of complementary mechanisms [212]. A key property of polyphenols is their ability to intercalate into the lipid bilayer, thereby modifying membrane fluidity and reducing O_2_ diffusion [213,214]. This physical remodeling of the membrane microenvironment limits the mobility of reactive species, thus attenuating the propagation of lipid peroxidation and contributing to the preservation of membrane integrity under oxidative stress [215]. Beyond these membrane-specific effects, polyphenols exert profound molecular benefits that extend to the cytoskeleton and redox-sensitive membrane proteins, particularly band 3 [216]. In RBCs exposed to oxidative challenges, polyphenol-rich plant extracts, such as those derived from *Citrus australasica* (finger lime), *Citrus bergamia* (bergamot), and *Euterpe oleracea* (açaí berries), have been shown to significantly reduce intracellular ROS levels and restore the functional and structural organization of band 3 and its anchorage to the spectrin–ankyrin cytoskeletal complex. For instance, in RBCs from prediabetic individuals, treatment with finger lime juice extract improved band 3 function and normalized α- and β-spectrin distribution, mitigating redox-related structural alterations [195]. Similarly, bergamot extract effectively counteracted oxidative modifications by inhibiting band 3 hyperphosphorylation, thus preserving cell deformability and mechanical stability [217]. Parallel findings with Açaí extract confirmed its ability to preserve band 3–cytoskeleton interactions and suppress oxidative stress-induced structural decline, particularly in models of premature aging [67]. These phytochemicals also confer protection at the Hb level, particularly in the context of metabolic dysfunction. Both finger lime and açaí extracts were shown to prevent metHb formation and significantly reduce HbA1c levels in RBCs from prediabetic subjects or those exposed to D-Gal [195,218]. These effects underscore the ability of polyphenols to prevent both oxidative and glycative modifications of Hb, thereby preserving its O_2_-carrying capacity and redox balance. Such actions are especially valuable in pathological contexts, especially diabetes and inflammation, where endogenous antioxidant systems are often compromised. In addition to their direct antioxidant activity, citrus-derived extracts have been shown to modulate the intracellular redox environment by restoring the GSH/GSSG ratio and normalizing the activity of antioxidant enzymes such as CAT and glutathione reductase (GR) [195,217]. This dual mechanism (direct ROS scavenging coupled with the support of enzymatic defense systems) enhances the redox buffering capacity of RBCs. Further supporting this view, Remigante et al. demonstrated that açaí extract fully reversed oxidative stress-induced hyperphosphorylation of band 3 in D-Gal-treated RBCs, suggesting its ability to stabilize redox-sensitive phosphorylation cascades involved in membrane signaling and structural integrity [67]. These findings align with broader evidence that identifies band 3 as a central redox sensor in RBCs, whose phosphorylation status is tightly regulated by kinase/phosphatase dynamics and is acutely sensitive to oxidative perturbations. By mitigating oxidative triggers and stabilizing signaling pathways, polyphenols contribute not only to structural preservation but also to the maintenance of functional plasticity in RBCs [219,220]. Collectively, the multi-targeted actions of polyphenolic compounds, ranging from membrane stabilization and cytoskeletal integrity to hemoglobin protection and enzymatic modulation, underscore their therapeutic potential in enhancing RBC viability under oxidative and metabolic stress [221,222].

Hydroxytyrosol (HT) and its primary metabolite homovanillyl alcohol (HVA), two potent polyphenols derived from extra virgin olive oil, have emerged as effective antioxidants in counteracting RBC damage induced in the context of heavy metal exposure. In particular, both compounds demonstrated the capacity to prevent the formation of metHb in RBCs exposed to mercury, a highly reactive pro-oxidant known to disrupt Hb structure and impair O_2_-carrying capacity [174]. Beyond hemoglobin protection, HT and HVA were also shown to restore intracellular glutathione (GSH) levels and normalize the GSH/GSSG ratio in mercury-exposed RBCs, underscoring their ability to reinforce cytosolic redox buffering capacity [174]. This action is particularly significant in the early stages of oxidative stress, where GSH depletion often precedes lipid peroxidation, protein oxidation, and membrane instability. By directly replenishing or preserving reduced GSH pools, these polyphenols help to prevent the cascade of molecular events that lead to irreversible oxidative damage. Collectively, the ability of HT and HVA to stabilize both Hb and the cytosolic redox environment highlights their dual role in protecting RBCs from functional decline under toxicological stress, with potential implications for dietary strategies aimed at enhancing systemic antioxidant defense.

Among the natural antioxidants evaluated, quercetin stands out due to its multifaceted biochemical properties and broad-spectrum efficacy. As a representative flavonoid, quercetin exhibits strong radical-scavenging and metal-chelating abilities, which effectively inhibit both initiation and propagation of lipid peroxidation within RBC membranes [35]. Beyond its role in preserving membrane lipid integrity, quercetin protects membrane proteins by attenuating glycation and oxidative post-translational modifications, primarily through the direct scavenging of reactive carbonyl species and the sequestration of redox-active metal ions. This feature limits AGEs formation and contributes to maintaining the structural and functional integrity of key proteins such as band 3 [35]. In the H_2_O_2_-induced oxidative stress model, quercetin reduces intracellular ROS levels and prevents oxidative and phosphorylative modification of band 3, thereby preserving cytoskeletal architecture, cell deformability, and overall membrane function [35]. Its anti-glycation potential has also been confirmed in models of premature aging, such as D-Gal-induced RBC damage, where quercetin fully prevented HbA1c formation, suggesting that it interferes with early glycation events at the Hb level [223]. Similar results were observed with polyphenol-rich açaí extract, which inhibited HbA1c formation at both low and moderate concentrations, further reinforcing the role of flavonoid-rich compounds in counteracting age-related Hb modifications [218]. Importantly, quercetin modulates redox-sensitive signaling pathways. In particular, it has been shown to preserve phosphorylation-dependent regulatory mechanisms involving band 3, a substrate of the tyrosine kinase Syk. In oxidative conditions, quercetin pre-treatment prevented band 3 hyperphosphorylation and functional inactivation while also reducing metHb formation, indicating a preserved signaling environment at the membrane–cytoskeleton interface [35]. In aging models, quercetin protected band 3-mediated anion exchange and prevented cytoskeleton structural deterioration [223]. However, the efficacy of quercetin is context dependent. Under conditions of depleted intracellular GSH, its oxidized form (quercetin-quinone) can interact with GSH to form adducts, potentially depleting the cellular thiol pool and exerting pro-oxidant effects. This phenomenon, known as the “quercetin paradox”, highlights the critical influence of intracellular antioxidant capacity in modulating the overall effect of quercetin and structurally related flavonoids [224]. Taken together, the evidence positions quercetin as a potent yet redox-sensitive molecule capable of targeting multiple oxidative stress pathways in RBCs, offering both structural and functional protection when supported by an adequate endogenous antioxidant environment.

Melatonin has emerged as a highly versatile antioxidant with the ability to protect RBCs through a combination of membrane-associated and cytosolic mechanisms. Its amphiphilic nature allows it to localize within both the lipid bilayer and the aqueous cytosolic environment, enabling broad-spectrum antioxidant protection. A key mechanism involves direct ROS scavenging, including OH^•^, thereby interrupting peroxidative chain reactions and preserving membrane lipid structure [225]. By stabilizing the lipid bilayer, melatonin not only prevents lipid peroxidation but also maintains membrane fluidity and inhibits the aggregation of oxidized membrane proteins, processes that are critical for RBC deformability and survival [225]. Beyond its membrane-level effects, melatonin also demonstrates significant cytosolic antioxidant functions. It contributes to redox homeostasis by modulating GSH recycling enzymes, such as G6PD and GR. In oxidative stress conditions, melatonin has been shown to preserve the intracellular GSH/GSSG ratio and sustain band 3 protein functionality, even when CAT activity is inhibited, indicating an operative independence from enzymatic catalysis [225]. This ability to maintain redox balance helps prevent early molecular events that typically lead to membrane and cytoskeletal damage. Additionally, melatonin is efficiently absorbed into RBCs under oxidative conditions, where it exerts a protective effect against protein carbonylation and Hb denaturation. Interestingly, such beneficial effect does not rely on direct replenishment of GSH, suggesting that melatonin can stabilize protein structures and delay oxidative modifications through alternative mechanisms. This includes the preservation of the conformational state of proteins such as band 3 and the delay of phosphorylation events that would otherwise impair cytoskeletal integrity and RBC signaling dynamics [226]. Anthocyanin-rich extracts, particularly those derived from *Callistemon citrinus*, have demonstrated potent antioxidant effects in RBCs, offering protection across multiple molecular targets involved in oxidative damage. These polyphenolic compounds significantly reduce intracellular ROS accumulation and lipid peroxidation markers, effectively preserving RBC morphology and membrane integrity under oxidative conditions [148]. By localizing within the lipid bilayer and interacting with membrane-associated proteins, anthocyanins help to stabilize structural components and prevent the progression of oxidative stress-induced damage.

One of the key findings related to *Callistemon citrinus* extracts is their ability to prevent band 3 protein aggregation and hyperphosphorylation in response to oxidative challenges, such as those induced by AAPH. These protective effects extend to the support of endogenous enzymatic defenses, highlighting the extract’s capacity to act as direct ROS scavengers, as well as a modulator of intracellular redox pathways [148]. The anthocyanin-enriched fraction has also shown efficacy in reducing metHb formation and in preventing the band 3 clustering, both hallmarks of RBC oxidative damage and early senescence. This dual action underscores the critical interplay between cytosolic and membrane components in maintaining RBC functionality under stress conditions [148]. Importantly, anthocyanin treatment fully restored intracellular GSH levels in AAPH-treated RBCs, suggesting that these compounds also bolster the cytosolic antioxidant capacity. As a consequence, anthocyanins prevent the compensatory overactivation of antioxidant enzymes such as CAT and SOD, thereby avoiding enzymatic exhaustion and contributing to long-term redox homeostasis [148]. This multifaceted activity positions anthocyanin-rich extracts as highly effective agents in the preservation of RBC structure and function.

Vitamin E (α-tocopherol) plays a central role in interrupting lipid peroxidation chain reactions within RBC membranes, acting as a lipid-soluble antioxidant that directly neutralizes lipid peroxyl radicals [227]. At the molecular level, α-tocopherol transfers a hydrogen atom to lipid peroxyl radicals, converting them into non-radical, more stable species [228]. In this process, it is itself oxidized to a tocopheryl radical, which is relatively unreactive and can be efficiently recycled by hydrophilic antioxidants such as vitamin C or GSH [229]. This recycling mechanism sustains the antioxidant capacity of vitamin E and ensures ongoing protection against oxidative lipid damage, which is critical to maintain membrane integrity and fluidity in RBCs exposed to ROS [230]. In synergy with vitamin E, vitamin C and N-acetylcysteine (NAC) exert complementary antioxidant effects by restoring reduced thiol groups in oxidized proteins, thereby preventing the formation of high-molecular-weight disulfide-linked aggregates [231,232]. This thiol protective activity is particularly relevant in preserving the function of membrane and cytoskeletal proteins like band 3, which are prone to oxidative cross-linking under stress conditions [233]. Furthermore, vitamin C contributes to redox homeostasis by inhibiting tyrosine nitration by directly scavenging ONOO^−^ or limiting the availability of its precursors [234,235]. Thus, the coordinated actions of vitamin E and its synergistic antioxidants form a robust molecular network that mitigates oxidative and nitrative injury at multiple biochemical levels, preserving RBC structure, protein functionality, and overall viability in oxidative environments [229,236,237].

## 6. Conclusions and Remarks

Despite lacking a nucleus and mitochondria, RBCs possess a finely regulated redox balance essential to their structural and functional integrity. As highlighted in this review, oxidative stress profoundly compromises key RBC molecular components, including membrane lipids and proteins, the cytoskeleton, and the intracellular environment. We have comprehensively described the central role of ROS-sensitive targets, particularly band 3, spectrin, and Hb, emphasizing the multifaceted and cooperative nature of oxidative damage and, concurrently, the univocal, often dysfunctional, cellular response. Oxidative alterations range from the trigger of chemical reactions mediated by reactive species to the activation of phosphorylation pathways. The main consequence of these oxidative insults is the impairment of rheological properties, as well as the alteration of O_2_ transport capacity, factors that are particularly critical in pathological conditions such as diabetes, chronic inflammation, and aging. On the other hand, we have shown how antioxidant molecules, including polyphenols, flavonoids, anthocyanins, melatonin, and vitamins C and E, exert their action at multiple molecular levels to prevent or reverse oxidative damage. These compounds, through direct ROS scavenging activity or as support for endogenous defense, maintain the reduced state of membrane, cytoskeletal, and cytoplasmic components, thus providing an overall protection of the peculiar functions of RBCs.

Taken together, these findings highlight the crucial relevance of redox homeostasis in RBC physiology and pathology. The pleiotropic actions of natural antioxidants point to their potential as therapeutic agents in preserving RBC integrity and delaying oxidative decline. Future research should aim to define optimal antioxidant combinations, delivery methods, and clinical applications to fully exploit their protective potential in conditions associated with systemic oxidative stress.

## Figures and Tables

**Figure 1 cimb-47-00655-f001:**
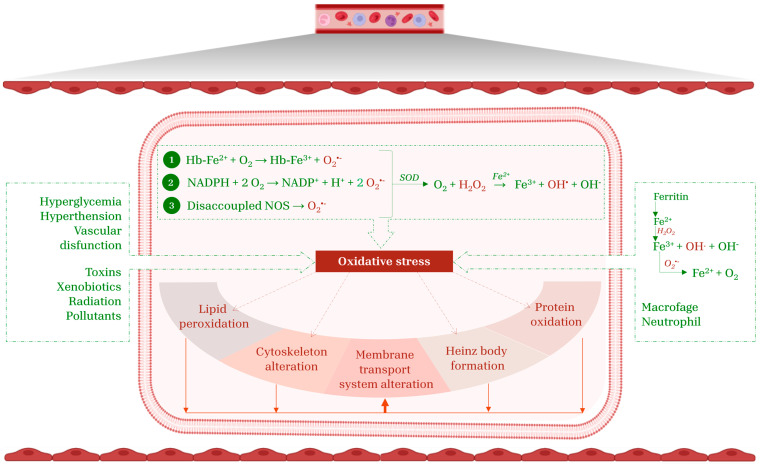
RBC targets of oxidative stress. The main endogenous mechanisms that promote oxidative stress include Hb auto-oxidation (1), NADPH oxidase activation (2), and NOS uncoupling (3). Exogenous sources of ROS originate from xenobiotics, pathological conditions, iron release from ferritin, as well as the activity of macrophages and neutrophils. Collectively, these factors establish a pro-oxidative environment that compromise RBC integrity, thus triggering lipid peroxidation, protein oxidation, alteration of the cytoskeleton and membrane transport systems, and the formation of Heinz bodies. This figure was created using BioRender.com. The colors in the cartoon are used solely for visual representation.

**Figure 2 cimb-47-00655-f002:**
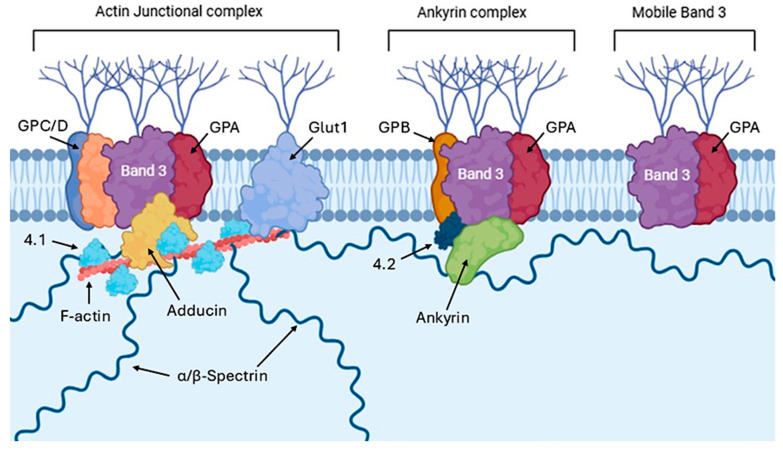
Schematic representation of the membrane–cytoskeleton interactions in RBCs, highlighting the actin junctional complex, ankyrin complex, and mobile band 3. This figure was created using BioRender.com.

**Figure 3 cimb-47-00655-f003:**
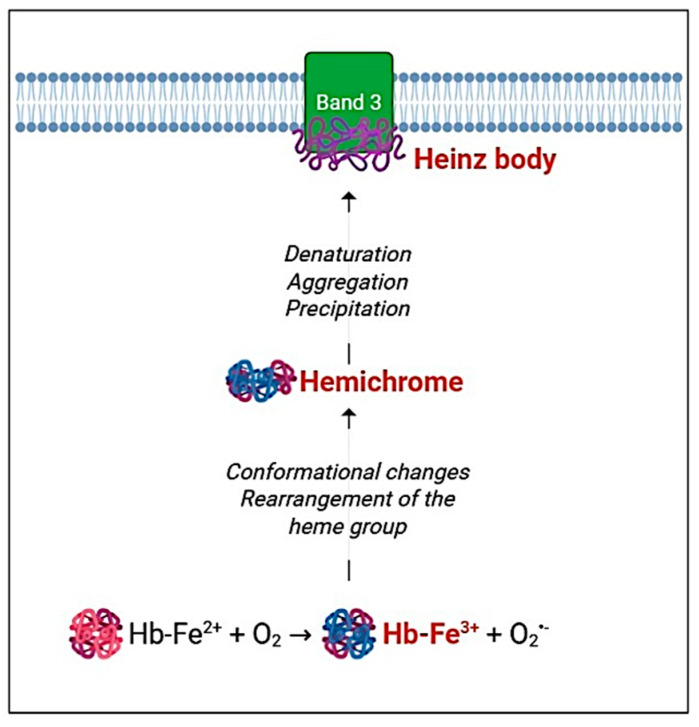
Heinz body formation. Oxidative stress induces hemoglobin oxidation (metHb), which undergoes conformational changes and heme group rearrangement. These oxidative alterations lead to the formation of hemichromes [184]. Within the cytosol, hemichromes denature, aggregate, and/or precipitate, subsequently binding to the cytosolic domain of band 3 and forming Heinz bodies [183]. This figure was created using BioRender.com. The colors in the cartoon are used solely for visual representation.

**Figure 4 cimb-47-00655-f004:**
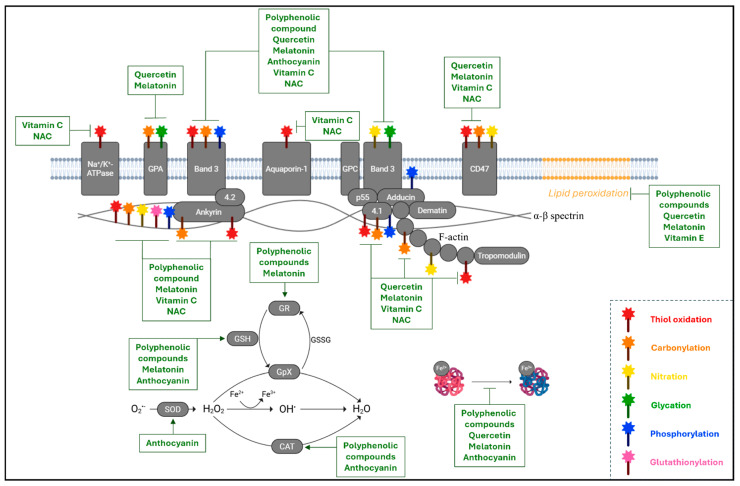
Oxidative alterations in RBCs and the beneficial effect of antioxidants. The image provides an overview of the oxidative changes in the main components of RBCs at the membrane, cytoskeleton, and intracellular levels. They consist of thiol oxidation (red wand), carbonylation (orange wand), nitration (yellow wand), glycation (green wand), phosphorylation (blue wand), and glutathionylation (pink wand). In parallel, the targets of the antioxidant molecules analyzed in this review have been highlighted either as inhibitors of the oxidative pathway or as boosters of the endogenous antioxidant defense system (catalase (CAT), superoxide dismutase (SOD), reduced glutathione (GSH), and glutathione reductase (GR)). GPA: glycophorin A; GPC: glycophorin C; GpX: glutathione peroxidase; GSSG: oxidized glutathione; NAC: N-acetylcysteine. This figure was created using BioRender.com.

**Table 1 cimb-47-00655-t001:** Major molecular targets of oxidative stress in human RBCs. Each target (membrane lipids, band 3 protein, spectrin–ankyrin complex, hemoglobin, Na^+^/K^+^-ATPase pump, and glycophorin A) corresponds to oxidative changes, which induce an effect at the cellular level, resulting in functional pathophysiological consequences.

Target Molecules and Structure	Oxidative Modification	Effect	Functional Consequence	References
Membrane lipids (PUFAs)	Lipid peroxidation	Decreased fluidity	Increased hemolysis susceptibility; altered ion permeability	[64,65]
Band 3 protein	Carbonylation and oxidation cross-links	Oligomerization and clustering	Impaired Cl^−^/HCO_3_^−^ exchange; loss of cytoskeletal anchoring	[65,66,67]
Spectrin–ankyrin complex	Cysteine and tyrosine oxidation	Weakened α/β-spectrin association	Decreased membrane elasticity; shape instability	[56]
Hemoglobin	Fe^2+^ → Fe^3+^ + heme dissociation	MetHb binds oxidized band 3	Band 3 clustering; micro-vesiculation production	[68,69]
Na^+^/K^+^-ATPase pump	Nitrosylation and carbonylation	Enzymatic inhibition or activation	Cell swelling; increased osmotic fragility	[70,71,72]
Glycophorin A (GPA)	Binding of e-amino groups of lysine residues with aldehyde species	Protein instability	Lower; susceptible to proteolysis	[73]
Aquaporin-1	Cysteine oxidation	Altered water channel conductance	Osmotic dysregulation	[74]

**Table 2 cimb-47-00655-t002:** Changes in RBC morphology in accordance with the oxidative pathway involved. The schematic representation highlights the effect of the oxidative pathway on the shape and function of human RBCs.

Abnormal RBC Shape	Oxidative Pathway	Morphological Effect	Pathophysiological Effect	References
Acanthocyte	Lipid peroxidation; membrane protein oxidation (B3p)	Irregular and asymmetrical spicule	Decreased deformability	[68,148,149]
Echinocyte	Lipid peroxidation; Metabolic disfunctions (decrease of ATP and NADPH content)	Regular and symmetrical spicule	Increased viscosity	[150,151]
Leptocyte	Lipid peroxidation; cytoskeletal protein oxidation (namely spectrin and ankyrin)	Increased area/volume ratio, flat and thin shape	Decreased O_2_ transport efficiency; increased osmotic fragility	[152]
Schistocyte	Lipid peroxidation; membrane and cytoskeletal oxidation (namely, spectrin, actin and band 3 protein)	RBC fragmentation	Decreased cell survival	[26,153]
Spherocyte	Membrane and cytoskeletal oxidation (namely, spectrin, actin, and band 3 protein); lipid peroxidation (subsequent process)	Loss of biconcave shape, spherical	Decreased deformability	[154]
Stomatocyte	Lipid peroxidation; membrane and cytoskeletal oxidation (namely, ion pump, and spectrin)	Central mouth-shaped pale area, cell swelling	Susceptibility to hemolysis	[155]

## Data Availability

No datasets were generated or analyzed during the current study.

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
