# Peer review of "Redox Homeostasis in Red Blood Cells: From Molecular Mechanisms to Antioxidant Strategies"

_cimb, 2025, doi:10.3390/cimb47080655_

Round 1
Reviewer 1 Report
Comments and Suggestions for Authors
RBCs are cells that need to deal with oxidative stress to maintain their homeostasis. The review addresses the impact of oxidative stress on cell membranes, detailing the mechanisms of lipid peroxidation and protein oxidation. Subsequently, the authors address the impacts on the cytoskeleton and cytosolic components, such as Hb. From there, the review details the cells' antioxidant mechanisms and, finally, compounds that complement mechanisms in protecting RBCs against oxidative stress. The topic is an important field of study, well explored by the authors in this review. However, several changes need to be made to improve the manuscript. Technical issues 1. First, I would like to make it clear that reviewing a manuscript without line numbers is complicated, which makes reviewing difficult. 2. Some topics are numbered, while others are not. For example, "2.1. Structure and functions of RBC membrane" and "2.3. Protein oxidation" are listed, but the topic that should be number 2, "Oxidative Stress at Plasma Membrane Level: Molecular Targets and Underlying Mechanisms," is not numbered. The same applies to the Introduction. 3. Sections 2.1 and 2.3 lack paragraphs to separate different topics, which makes reading difficult. Although the remaining topics have paragraphs, some take up practically a whole page. This needs urgent revision. 4. Authors should include an abbreviations section. Figures and Tables 1. In Fig. 1, correct the term pollutants and make clear the internal and external effects. In addition, wouldn't changes in the cytoskeleton and membrane transport system be consequences of the oxidized protein? 2. In Fig. 2, what is mobile band 3? This needs to be explained in the text. 3. In Fig. 4, CAT, SOD, GSH and GR need full names in figure caption. In addition, NAC (N-Acetylcysteine) must be defined in the manuscript and in the Figure caption. 4. GPA and Aquaporin-1 appear in Figure 4 but are not highlighted in Table 1. What is the reason for this difference? 5. Table 2. In Pathophysiological effect, the authors use ";" to separate two processes. This can be used in the "Oxidative pathway" column.
Suggestions 1. "Keywords: RBCs; oxidative stress; Reactive Oxygen Species (ROS); rheological properties; antioxidants". If ROS appears as the full name, RBC should as well. In addition, the term "rheological properties" appears only three times throughout the manuscript. Should it be considered a keyword? 2. "is regulated by O2 partial pressure, pH and 2,3 diphosphoglycerate levels [3]." and " thereby promoting glycolysis and 2,3-bisphosphoglycerate synthesis via the Rapoport-Luebering shunt". Authors should choose a nomenclature for these two times the term appears in the manuscript. 3. These two references were not entered as numbers: "from reduced forms (oxyHb and deoxyHb) to oxidized one (methemoglobin (metHb)) 10.3390/antiox12091736]." and "morphology and membrane integrity under oxidative conditions [10.3389/fphys.2023.1303815]." 4. "Table 37. The lipid composition of RBC membrane represents one of the most intriguing and fun". I didn't understand the reference to Table 37. 5. "particularly glycophorin A (GPA), are prominent sialoglycoproteins. Glycophorin A". The acronym had already been defined previously. Furthermore, the authors repeat the full name several times. 6. "Functional Role Of Cytosolic Components In Rbc Response To Oxidative Stress" and " Multilevel Protective Roles Of Natural Antioxidants In Preserving Rbc Structure And Function Under Oxidative Stress". RBC must be capitalized and topics must be numbered. 7. "Despite the lack of nuclei and mito-chondria, mature RBCs retain sufficient cytosolic enzymatic activity for partial operation of the Lands cycle, depending on the availability of ATP, coenzyme A and acyl-CoA derivatives [55]." The authors could cite an article that demonstrates this, and not just the review that addressed this topic.
8. "high-abundance integral proteins such as band 3, CD47, and glycophorin A, which are". The abbreviation of glycophorin A should be used. 9. "In diabetic patients, the accumulation of these Hb degradation products can severely impair tissue oxygenation, further exacerbating disease complications [177]." This sentence was separated into a single paragraph. It needs to be integrated into the previous paragraph. 10. "Conclusions And Remarkers" change to "Conclusions And Remarks".

Author Response
Dear Editor,
thank you for considering our manuscript for a possible publication in Current Issues in Molecular Biology (MDPI). We hereby provide a point-by-point response to each comment of the reviewers.
Reviewer 1
RBCs are cells that need to deal with oxidative stress to maintain their homeostasis. The review addresses the impact of oxidative stress on cell membranes, detailing the mechanisms of lipid peroxidation and protein oxidation. Subsequently, the authors address the impacts on the cytoskeleton and cytosolic components, such as Hb. From there, the review details the cells' antioxidant mechanisms and, finally, compounds that complement mechanisms in protecting RBCs against oxidative stress. The topic is an important field of study, well explored by the authors in this review. However, several changes need to be made to improve the manuscript.
Technical issues.
First, I would like to make it clear that reviewing a manuscript without line numbers is complicated, which makes reviewing difficult. Thanks to the reviewer for the suggestion. Done.
Some topics are numbered, while others are not. For example, "2.1. Structure and functions of RBC membrane" and "2.3. Protein oxidation" are listed, but the topic that should be number 2, "Oxidative Stress at Plasma Membrane Level: Molecular Targets and Underlying Mechanisms," is not numbered. The same applies to the Introduction. 3. Sections 2.1 and 2.3 lack paragraphs to separate different topics, which makes reading difficult. Although the remaining topics have paragraphs, some take up practically a whole page. This needs urgent revision. Thanks to the reviewer for the suggestion. Done.
Authors should include an abbreviations section.
Thanks to the reviewer for the suggestion. Done.
Figures and Tables.
In Fig. 1, correct the term pollutants and make clear the internal and external effects. In addition, wouldn't changes in the cytoskeleton and membrane transport system be consequences of the oxidized protein? Thanks to the reviewer for the suggestions. The term “pollutans” has been corrected. This image is designed to highlight the general effects of oxidative stress (generated by exogenous and endogenous sources) on red blood cells, in order to introduce the topic to the reader. The specific effects on individual components are discussed in detail in the following sections. As regards the alteration of membrane transport systems, this is a consequence of several factors, as highlighted in the following sections. Protein oxidation undoubtedly plays a central role, as well as lipid peroxidation, cytoskeletal oxidative modifications and Heinz body formation [10.1021/bi400405p; 10.1016/s0891-5849(99)00149-5; 10.1155/2013/985210; 10.1172/JCI112696.]. Therefore, the image has been modified to clarify these mechanisms.
In Fig. 2, what is mobile band 3? This needs to be explained in the text. Thanks to the reviewer for the suggestion. The following sentence has been added to the text (section 3.1): “In addition to its association with the ankyrin and junctional complexes, approximately 27% of Band 3 is unanchored to the cytoskeleton and therefore free to diffuse laterally in the membrane [10.1074/jbc.M111.294439; 10.1073/pnas.77.5.2537]”.
In Fig. 4, CAT, SOD, GSH and GR need full names in figure caption. In addition, NAC (N-Acetylcysteine) must be defined in the manuscript and in the Figure caption. Thanks to the reviewer for the suggestion. Done.
GPA and Aquaporin-1 appear in Figure 4 but are not highlighted in Table 1. What is the reason for this difference? Thanks to the reviewer for the suggestion. It was a typo. They have been added to the table.
Table 2. In Pathophysiological effect, the authors use ";" to separate two processes. This can be used in the "Oxidative pathway" column. Thanks to the reviewer for the suggestion. Done.
Suggestions.
"Keywords: RBCs; oxidative stress; Reactive Oxygen Species (ROS); rheological properties; antioxidants". If ROS appears as the full name, RBC should as well. In addition, the term "rheological properties" appears only three times throughout the manuscript. Should it be considered a keyword? Thanks to the reviewer for the suggestion. “RBCs” has been written in full, and the keyword “rheological properties” has been replaced by “Band 3 protein; phosphorylation pathways; RBC cytoskeleton; RBC membrane; hemoglobin (Hb)”.
"is regulated by O2 partial pressure, pH and 2,3 diphosphoglycerate levels [3]." and " thereby promoting glycolysis and 2,3-bisphosphoglycerate synthesis via the Rapoport-Luebering shunt". Authors should choose a nomenclature for these two times the term appears in the manuscript. Thanks to the reviewer for the suggestion. The nomenclature “2,3-biphosphoglycerate” has been chosen.
These two references were not entered as numbers: "from reduced forms (oxyHb and deoxyHb) to oxidized one (methemoglobin (metHb)) 10.3390/antiox12091736]." and "morphology and membrane integrity under oxidative conditions [10.3389/fphys.2023.1303815]." Thanks to the reviewer for the suggestion. Done.
"Table 37. The lipid composition of RBC membrane represents one of the most intriguing and fun". I didn't understand the reference to Table 37. Thanks to the reviewer for the suggestion. It was a typo. The formatting has been corrected.
"Particularly glycophorin A (GPA), are prominent sialoglycoproteins. Glycophorin A". The acronym had already been defined previously. Furthermore, the authors repeat the full name several times. Thanks to the reviewer for the suggestion. Done.
"Functional Role Of Cytosolic Components In Rbc Response To Oxidative Stress" and " Multilevel Protective Roles Of Natural Antioxidants In Preserving Rbc Structure And Function Under Oxidative Stress". RBC must be capitalized and topics must be numbered. Thanks to the reviewer for the suggestion. Done.
"Despite the lack of nuclei and mitochondria, mature RBCs retain sufficient cytosolic enzymatic activity for partial operation of the Lands cycle, depending on the availability of ATP, coenzyme A and acyl-CoA derivatives [55]." The authors could cite an article that demonstrates this, and not just the review that addressed this topic. Thanks to the reviewer for the suggestion. The reference “Wu, H., Bogdanov, M., Zhang, Y. et al. Hypoxia-mediated impaired erythrocyte Lands’ Cycle is pathogenic for sickle cell disease. Sci Rep 6, 29637 (2016). https://doi.org/10.1038/srep29637” has been added.
"High-abundance integral proteins such as band 3, CD47, and glycophorin A, which are". The abbreviation of glycophorin A should be used. Thanks to the reviewer for the suggestion. Done.
"In diabetic patients, the accumulation of these Hb degradation products can severely impair tissue oxygenation, further exacerbating disease complications [177]." This sentence was separated into a single paragraph. It needs to be integrated into the previous paragraph. Thanks to the reviewer for the suggestion. Done.
"Conclusions and Remarkers" change to "Conclusions and Remarks". Thanks to the reviewer for the suggestion. Done
Reviewer 2 Report
Comments and Suggestions for Authors
Title: REDOX HOMEOSTASIS IN RED BLOOD CELLS: FROM MOLECULAR MECHANISMS TO ANTIOXIDANT STRATEGIES.
In this paper, the authors study oxidative stress in red blood cells and the redox regulation required to maintain membrane integrity, cytoskeletal organization, and metabolic function. Disruption of these processes leads to a series of oxidative events that lead to premature red blood cell removal. The authors conclude that oxidative stress is at the core of red blood cell dysfunction in both physiological and pathological aging.
Abstract: The conclusion reported in this article does not accurately reflect the significance of the research conducted. Please supplement this paragraph.
This paper describes something new, which could be interesting; however, I have some concerns.
This paper seems good to me, informative and adds something new to the current literature.
This article could be better introduced with a little cellular biology.
The tables only have the title without a clear legend explaining the data.
The figures should explain the reported content.
MAPK plays an important role in the stress response, including in red blood cells. Oxidative stress can activate MAPKs via upstream kinases, phosphorylating intracellular targets for stress protection, cell cycle modulation in erythroid precursors, and apoptosis. In the light of these concepts, to make this paper more interesting for the readers of this important journal, the authors should expand a bit the discussion (or introduction). Below I report an interesting article that should be studied, incorporate the meaning and report it briefly in the discussion and in the list of references.
Saggini R, Pellegrino R. MAPK is implicated in sepsis, immunity, and inflammation. International Journal of Infection. 2024;8(3):100-104. (www.biolife-publisher.it).
In addition, Redox homeostasis in red blood cells and the PI3K/Akt/mTOR signaling pathway is a very interesting context in which intracellular signaling and oxidative stress influence each other, even in erythrocytes, and especially in their erythroid precursors. Again, here, we report an article which has been recently published that should be studied, incorporate the meaning and report briefly in the discussion or introduction, and in the list of the references.
Avivar-Valderas A. Inhibition of PI3Kβ and mTOR influence the immune response and the defense mechanism against pathogens. International Journal of Infection. 2023;7(2):46-49. (www.biolife-publisher.it).
I believe these suggestions are important for improving this paper. Without these corrections the paper cannot be published. So I recommend minor revision.
Comments on the Quality of English LanguageThe English could be improved
Author Response
Dear Editor,
thank you for considering our manuscript for a possible publication in Current Issues in Molecular Biology (MDPI). We hereby provide a point-by-point response to each comment of the reviewers.
Reviewer 2
In this paper, the authors study oxidative stress in red blood cells and the redox regulation required to maintain membrane integrity, cytoskeletal organization, and metabolic function. Disruption of these processes leads to a series of oxidative events that lead to premature red blood cell removal. The authors conclude that oxidative stress is at the core of red blood cell dysfunction in both physiological and pathological aging.
Abstract: The conclusion reported in this article does not accurately reflect the significance of the research conducted. Please supplement this paragraph. Thanks to the reviewer for the suggestion. The following sentences are be added to the text: Redox regulatory mechanisms in RBCs are required to maintain membrane integrity, cytoskeletal organization and metabolic function. Disruption of these processes causes several oxidative processes that trigger RBC premature removal.
This paper describes something new, which could be interesting; however, I have some concerns.
This paper seems good to me, informative and adds something new to the current literature. Thanks to the reviewer for this opinion.
This article could be better introduced with a little cellular biology. Thanks to the reviewer for the suggestion. In order to avoid repetition due to the abundance of documents already available in the scientific literature, the authors have included solely the information necessary for the context of this review.
The tables only have the title without a clear legend explaining the data. Thanks to the reviewer for the suggestion. Captions are modified in accordance with reviewer suggestion.
The figures should explain the reported content. Thanks to the reviewer for the suggestion. The caption for Figure 4 has been implemented.
MAPK plays an important role in the stress response, including in red blood cells. Oxidative stress can activate MAPKs via upstream kinases, phosphorylating intracellular targets for stress protection, cell cycle modulation in erythroid precursors, and apoptosis. In the light of these concepts, to make this paper more interesting for the readers of this important journal, the authors should expand a bit the discussion (or introduction). Below I report an interesting article that should be studied, incorporate the meaning and report it briefly in the discussion and in the list of references. Saggini R, Pellegrino R. MAPK is implicated in sepsis, immunity, and inflammation. International Journal of Infection. 2024;8(3):100-104. (www.biolife-publisher.it). Thanks to the reviewer for the suggestion. The reference has been added.
In addition, Redox homeostasis in red blood cells and the PI3K/Akt/mTOR signaling pathway is a very interesting context in which intracellular signaling and oxidative stress influence each other, even in erythrocytes, and especially in their erythroid precursors. Again, here, we report an article which has been recently published that should be studied, incorporate the meaning and report briefly in the discussion or introduction, and in the list of the references. Avivar-Valderas A. Inhibition of PI3Kβ and mTOR influence the immune response and the defense mechanism against pathogens. International Journal of Infection. 2023;7(2):46-49. (www.biolife-publisher.it). We thank the reviewer for raising this point. The aim of this review is to highlight oxidative changes in mature red blood cells. Therefore, although interesting, the PI3K/Akt/mTOR signaling pathway in erythroid precursors is beyond the scope of this review.
I believe these suggestions are important for improving this paper. Without these corrections the paper cannot be published. So, I recommend minor revision.
Round 2
Reviewer 1 Report
Comments and Suggestions for Authors
The authors answered the questions and made the necessary modifications to improve the manuscript.
Author Response
We confirm that we used the service offered by MDPI to make the necessary English language modifications to improve the manuscript.